# A Novel Application on the Drive Elements of Using Electrical Contact Resistance and Friction Coefficient for Evaluating Induction Heat Treatment

**DOI:** 10.3390/ma14040865

**Published:** 2021-02-11

**Authors:** Yuh-Ping Chang, Hsiang-Yu Wang, Huann-Ming Chou

**Affiliations:** Green Energy Technology Research Center, Department of Mechanical Engineering, Kun Shan University, No.195, Kunda Rd., Yongkang Dist., Tainan City 710303, Taiwan; andysti486@yahoo.com.tw (H.-Y.W.); hmchou@mail.ksu.edu.tw (H.-M.C.)

**Keywords:** drive elements, induction hardening, wear, synchronous response

## Abstract

The load on drive elements under extreme pressure conditions is significantly larger than that used in machine tools. When operating under a heavy load for a long period, large deformation and severe wear between the ball and the track are more likely to occur. To reduce wear, the most fundamental solution is to improve the surface properties of the material. Moreover, heat treatment is the most effective method to improve the surface properties of materials, thereby achieving wear resistance and low friction. It is necessary to develop a new heat treatment technology for wear resistance in extreme pressure conditions. Therefore, this study conducted experiments using a reciprocating friction tester. The responses of electrical contact resistance and the friction coefficient were measured synchronously to investigate wear resistance and low friction of the alloy steels after the induction heat treatment. Then, the results were compared and verified with low-carbon alloy steel after the traditional carburizing heat treatment. The experimental results show that the application of new induction heat treatment technology can not only improve the performance of drive components, but also save time and energy, and streamline the production process of the drive components. Therefore, the results of these wear analyses confirm that the induction heat treatment mode can replace the traditional carburizing heat treatment mode for drive elements.

## 1. Introduction

Due to the advancement of science and technology, the fields of machine tools and precision machinery are moving towards high precision, high speed, high load, long life, and automation. Hence, there are considerable requirements for product quality, dimensional accuracy, and service life. At the present, drive elements play crucial roles in precision machinery, and they must have low-friction and antiwear properties [1,2,3]. Due to the popularity of electronically controlled drive elements and the application requirements for heavy-duty mechanical systems, fretting wear has attracted greater attention because it significantly affects fatigue life and the reliability of related mechanical systems [4,5,6,7]. The effective application of surface hardening has been used to evaluate the potential to address these problems [8,9,10]. To revitalize machinery-related industries, it is necessary to explore new heat treatment methods to develop low-friction and wear resistant drive elements [11].

After long-term operation of mechanical components, the problem of poor dimensional accuracy due to wear arises, and long-term accumulation causes mechanical damage and failure. As a result, the improvement of material surface engineering is an important topic to ensure the wear resistance life of drive elements [12,13]. Regarding the related research on improving the impact of surface engineering on wear, numerous methods have been investigated, such as polishing, coating, and heat treatment. These approaches can effectively achieve the purpose of improving surface properties [14,15,16]. Among these, proper heat treatment is the most effective, and can not only directly improve the surface hardness, but also change the structure of the substrate, so that the overall performance of the drive elements can be greatly improved [10,17].

After basic heat treatment, such as quenching and tempering, the overall material properties are improved [18]. Then, the most important surface hardening process is carried out. The traditional surface hardening treatment is the carburizing process, which has been widely used in industry. However, because the carburizing process must allow enough time for the carbon atoms to carry out sufficient thermal diffusion, the processing time can often be as long as ten hours, thus consuming electricity and time. Furthermore, the manufacturing cost is high. These outcomes are not in line with the “energy saving” and “resource saving” principles of green technology. The traditional surface hardening treatment includes carburizing for 12 h, quenching for 1 h, and tempering for 1.5 h, for a total time of about 14.5 h. The induction heat treatment includes quenching 1 h, tempering 1.5 h, and induction for only 20 s, for a total time of about 2.5 h. Therefore, power consumption is reduced by about 5.8 times.

Based on the author’s previous related results [19], a series of effects of various heat treatments on the wear performance of drive elements have been analyzed [20,21,22]. After a comprehensive evaluation, it was found that the induction heat treatment provides the opportunity to achieve simultaneous energy saving, time saving, and reliable performance. Therefore, this study selected the influence of induction heat treatment on wear performance of the drive elements as the research direction, with the aim of replacing the traditional carburizing heat treatment process.

## 2. Experimental Apparatus and Procedures

### 2.1. Experimental Apparatus

A reciprocating friction tester with measuring systems, shown in Figure 1, was used to study the wear properties of a SUJ2 ball sliding against different steels. A crank-slider mechanism was used to drive a ball specimen and the stroke of the crank was set to 6 mm. A cylinder specimen was placed on the stationary rest and connected to the load cell to measure the friction coefficient. A normal load of 100 N was related to the ball through the cylinder and disposed along the level rule. To remain in complete contact during the test process, a softer spring with an oil damper was employed in the loading system.

### 2.2. Experimental Specimens

The ball was made of the high-carbon chromium alloy steel (ϕ6.35 mm). Furthermore, the cylinder specimens were made of low-carbon alloy steel or high-carbon chromium alloy steel. The contents (wt %) are given in Table 1 and Table 2, respectively. The sizes and shapes of the test specimens are shown in Figure 2. The ball specimen was a commercial product of SUJ2, and the surface hardness was about HRC 66. Moreover, it was replaced with a new ball after each experiment. Quenching took about 1 h, and the appropriate range of quenching temperature was 820–860 °C. Tempering took about 1.5 h, and the tempering temperature was about 600–700 °C. When the tempering temperature increases by 10 °C, the surface hardness decreases by about HRC 1. To meet the requirements of workability, the hardness of the test specimen must be in the range of HRC 25 to 30. If it exceeds this range, the tool is extremely susceptible to damage during the processing of the spiral profile. Therefore, the improvement method adopted in this study is to appropriately increase the tempering temperature to reduce the hardness of the test piece. The detailed heat treatment process of the four kinds of cylinder specimens is shown in Table 3. The A test piece is the “traditional heat treatment mode” of low-carbon chromium alloy steel. The heat treatment time of carburization is about 12 h. The B1, B2, and B3 test pieces are the “online induction heat treatment mode” of high-carbon chromium alloy steel, and the heat treatment time of induction is only about 20 s.

### 2.3. Experimental Procedures and Conditions

The experimental conditions are shown in Table 4. The response times of the measuring systems were shorter than 1 ms, and the measurement accuracy was 0.1% of the overall measurement scale. The room temperature remained at 26 ± 2 °C, and the relative humidity remained at 65 ± 5%. Test specimens were cleaned with acetone in an ultrasonic cleaner before each experiment and securely locked in position. Then, grease was applied evenly to the cylinder specimen. The voltage signals of electrical contact resistance and friction coefficient were recorded by a data acquisition system during the tests, and then fed to a personal computer for data analysis. After the tests, the worn surfaces of the cylinder specimens were observed using an optical microscope. Moreover, the wear depth was measured by a surface roughness meter.

## 3. Results and Discussions

### 3.1. Dynamic Responses of Electrical Contact Resistance

Figure 3 shows the typical responses of electrical contact resistance for a ball sliding against the A specimen. It can be seen in the figure that electrical contact resistance varies from 0 to 4.5 kΩ. According to the results, the upper and lower test specimens were in good contact during the friction time of 0–30 s. Then, the vibration frequency of electrical contact resistance increased gradually, which means that serious wear was gradually generated between the interfaces. This occurred because the generation and exclusion of the abrasion particles was extremely fast, and the beatings of the contact interface were intense.

Figure 4 shows the variation of electrical contact resistance for a ball sliding against the B1 specimen. The electrical contact resistance varied from 0 to 0.2 kΩ. This indicates that the upper and lower test specimens were in good contact during the friction time of 0–30 s. Even when the friction time reached about 1 h, only slight wear was produced. Hence, the overall wear was significantly less than that of the low carbon alloy steel of the A specimen.

Figure 5 shows the case of the B2 specimen. The electrical contact resistance varied from 0 to 0.5 kΩ. The results indicate that mild wear occurred during the whole friction test. These changes are more obvious in the initial stage of friction. Similar to Figure 4, the overall wear was significantly less than that of the A specimen.

The variation of electrical contact resistance for the B3 specimen is shown in Figure 6. The electrical contact resistance varied from 0 to 1.9 kΩ; this result is similar to that of Figure 5, but with a larger value. The results still show the overall wear is less than that of the A specimen.

### 3.2. Dynamic Responses of Friction Coefficient

Figure 7 shows the typical responses of the friction coefficient for a ball sliding against the A specimen. It can be seen in Figure 7a that the friction coefficient was about 0.2 over the friction distance of 0–0.4 m. Then, the friction coefficient increased gradually to about 1.2 over the friction distance of 0.4–2.4 m. This indicates that the running-in period was about 0.4 m. Figure 7b shows the stable friction coefficient was about 2.5 over the friction distance of 288–290.4 m. From a comparison of the results of Figure 7 with those of Figure 3, when the friction coefficient gradually increased, the oscillation frequency of the electrical contact resistance also increased, and the wear was more severe. The average friction coefficient during the whole process was about 1.83.

Figure 8 shows the typical responses of the friction coefficient for a ball sliding against the B1 specimen. The friction coefficient was about 0.15 over the friction distance of 0–0.5 m. Then, the friction coefficient gradually increased to about 1.0 over the friction distance of 0.5–2.4 m. Figure 8b shows that the unstable friction coefficient was 0.5–1.3 over the friction distance of 288–290.4 m. The average friction coefficient during the whole process was about 0.88.

Figure 9 shows the case of the B2 specimen. The friction coefficient was only about 0.1 over the friction distance of 0–0.6 m. Then, the friction coefficient varied from 0.2 to 0.6 over the friction distance of 0.6–2.4 m. Figure 9b shows that the stable friction coefficient was about 0.35 during the friction distance of 288–290.4 m. The average friction coefficient during the whole process was about 0.55.

The variation of the friction coefficient for the B3 specimen is shown in Figure 10. The friction coefficient was only about 0.1 over the friction distance of 0–0.4 m. Then, the friction coefficient varied from 0.25 to 0.6 over the friction distance of 0.4–2.4 m. Figure 10b shows the stable friction coefficient was about 0.35 over the friction distance of 288–290.4 m. The average friction coefficient during the whole process was about 0.6.

The average friction coefficient of the four specimens is shown in Figure 11. This figure shows that all of the alloy steels after the induction heat treatment had lower friction coefficients than that of the low-carbon alloy steel, the current mass production base material of drive elements, after the traditional carburizing heat treatment. Moreover, the cryogenic treatment also had an obvious low-friction effect.

### 3.3. Optical Microscope Images of the Worn Surface

The optical microscopes images of the worn surface of the cylinder specimens after sliding the ball against the four cylinder materials are shown in Figure 12. All of the alloy steels after the induction heat treatment showed less abrasion than the low-carbon alloy steel after the traditional carburizing heat treatment. Moreover, the worn surface of the B2 test piece with the cryogenic treatment was smoother.

### 3.4. Quantitative Analysis of Wear Depth

The wear depths of the cylinder specimen for a ball sliding against the four cylinder materials are shown in Figure 13, Figure 14, Figure 15 and Figure 16. It can be seen from the figures that all of the alloy steels after the induction heat treatment showed more resistance to wear than the low-carbon alloy steel after the traditional carburizing heat treatment.

Based on the above experimental results of electrical contact resistance, friction coefficient, optical microscope images, and wear depth, it can be seen that the changes of the four characteristics correspond to each other. The greater the friction coefficient, the higher the oscillation frequency of the electrical contact resistance, and the faster the “generation” and “elimination” of wear particles, resulting in more serious wear.

For the accuracy of the experiment, all of the cylinder specimens were subjected to three reproducibility experiments. The average wear depth result is shown in Figure 17. This figure shows that the wear depth of the B2 test piece with the cryogenic treatment is obviously smaller. Hence, the effect of the cryogenic treatment on wear resistance is significant. In addition, the wear resistance of the B3 test piece with the higher temperature tempering is also acceptable. To preserve the tool life in long-term processing, the heat treatment process of the B3 specimen is most suitable to be used in industry. The original carbon content of the A specimen was only about 0.17–0.23 wt %, and the final surface hardness was HRC 59–61 after carburization for 12 h. The original carbon contents of B1, B2, and B3 were about 0.95–1.10 wt %. Following the quenching and tempering treatment, a final surface hardness of HRC 58–62 was obtained using 20 s of induction. This not only saves a significant amount of electrical energy, but also simplifies the manufacturing process of the transmission element production line due to the reduced heat treatment time, while also conforming to the development strategy of industrial automation. Applying the results of this research can not only improve the performance of drive elements, but also save time, energy, and production lines for the manufacturing process, and the production equipment of drive elements. Therefore, this research confirms that the online induction heat treatment mode after proper material processing can replace the nonproduction line-based traditional carburizing heat treatment mode.

## 4. Conclusions

The tribology properties of alloy steels after induction heat treatment and low-carbon alloy steel after traditional carburizing heat treatment were investigated and compared. The conclusions are as follows:The experimental results of electrical contact resistance, friction coefficient, optical microscope images, and wear depth showed correspondence to each other. These results allowed comprehensive evaluation of tribological performance.All of the alloy steels after induction heat treatment showed less abrasion than the low-carbon alloy steel after the traditional carburizing heat treatment. Moreover, better abrasion resistance was obtained by adding cryogenic treatment.To protect the tool life in long-term processing, the experimental results prove that the heat treatment process with higher temperature tempering is acceptable.The induction high-carbon chromium alloy steel can obtain the dual advantages of antiwear and low friction properties. This not only saves a significant amount of electrical energy, but also simplifies the manufacturing process of the transmission element production line due to the reduced heat treatment time.The online induction heat treatment mode after proper material processing can replace the nonproduction line-based traditional carburizing heat treatment mode.

## Figures and Tables

**Figure 1 materials-14-00865-f001:**
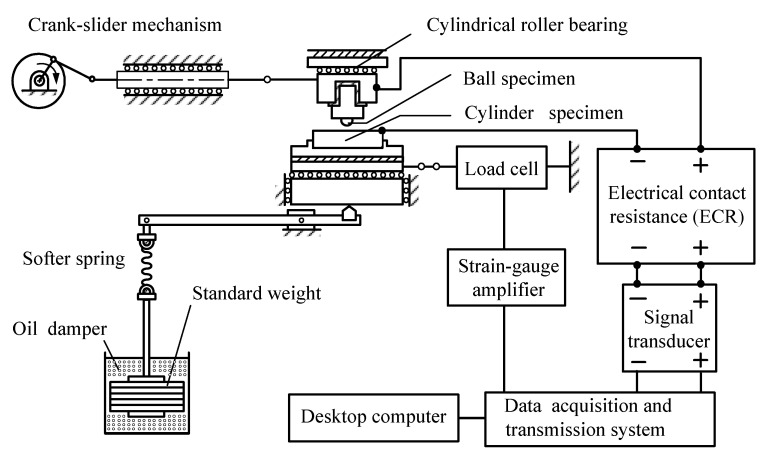
The schematic diagram of the reciprocating friction tester with the measuring systems.

**Figure 2 materials-14-00865-f002:**
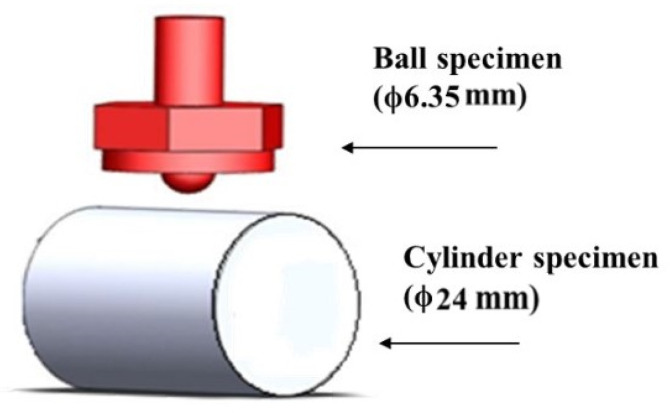
Schematic diagram of the overall experimental sample configuration.

**Figure 3 materials-14-00865-f003:**
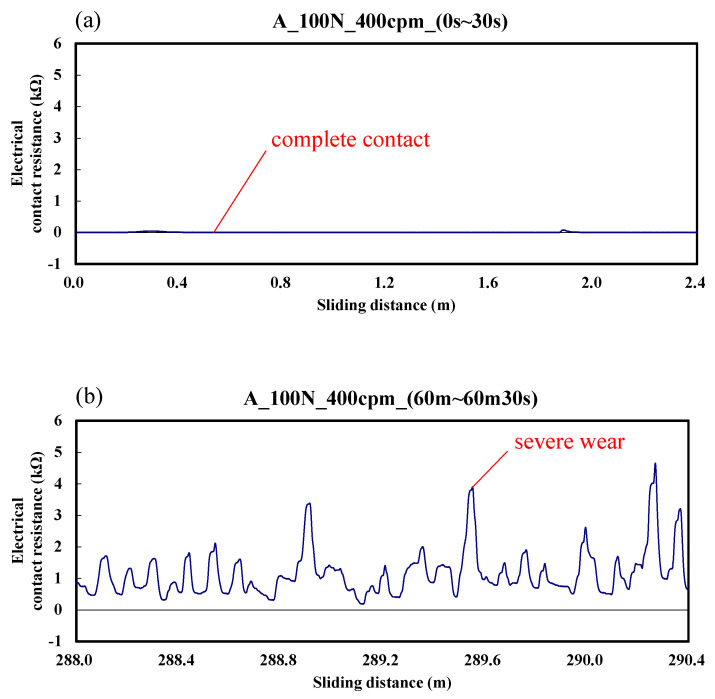
Dynamic responses of electrical contact resistance of the A specimen. (**a**) sliding distance of 0.0–2.4 m; (**b**) sliding distance of 288.0–290.4 m.

**Figure 4 materials-14-00865-f004:**
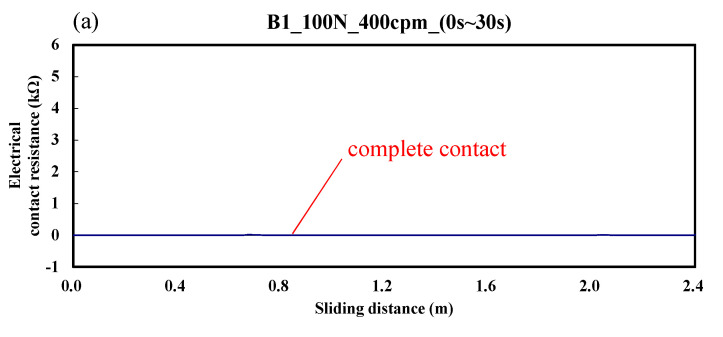
Dynamic responses of electrical contact resistance of the B1 specimen. (**a**) sliding distance of 0.0–2.4 m; (**b**) sliding distance of 288.0–290.4 m.

**Figure 5 materials-14-00865-f005:**
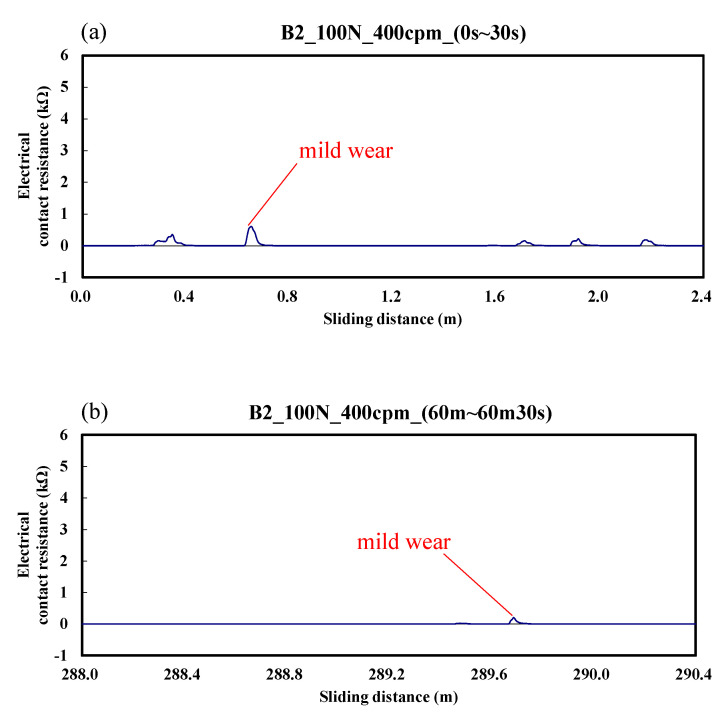
Dynamic responses of electrical contact resistance of the B2 specimen. (**a**) sliding distance of 0.0–2.4 m; (**b**) sliding distance of 288.0–290.4 m.

**Figure 6 materials-14-00865-f006:**
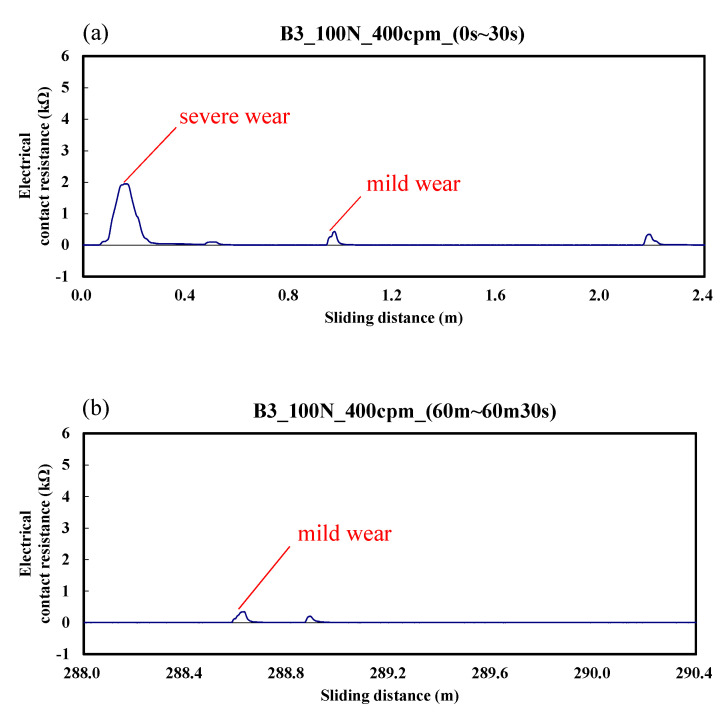
Dynamic responses of electrical contact resistance of the B3 specimen. (**a**) sliding distance of 0.0–2.4m; (**b**) sliding distance of 288.0–290.4 m.

**Figure 7 materials-14-00865-f007:**
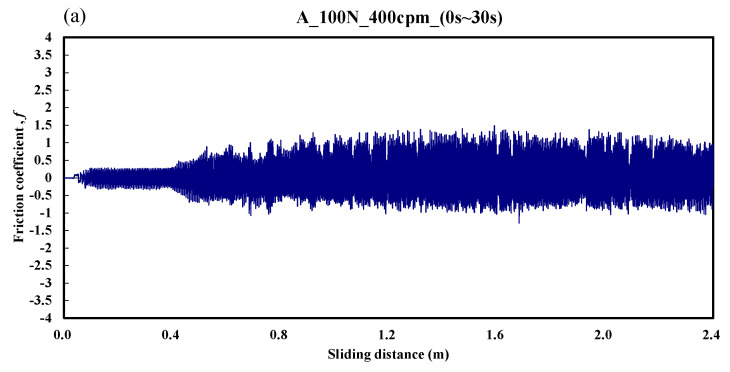
Dynamic responses of the friction coefficient of the A specimen. (**a**) sliding distance of 0.0–2.4m; (**b**) sliding distance of 288.0–290.4 m.

**Figure 8 materials-14-00865-f008:**
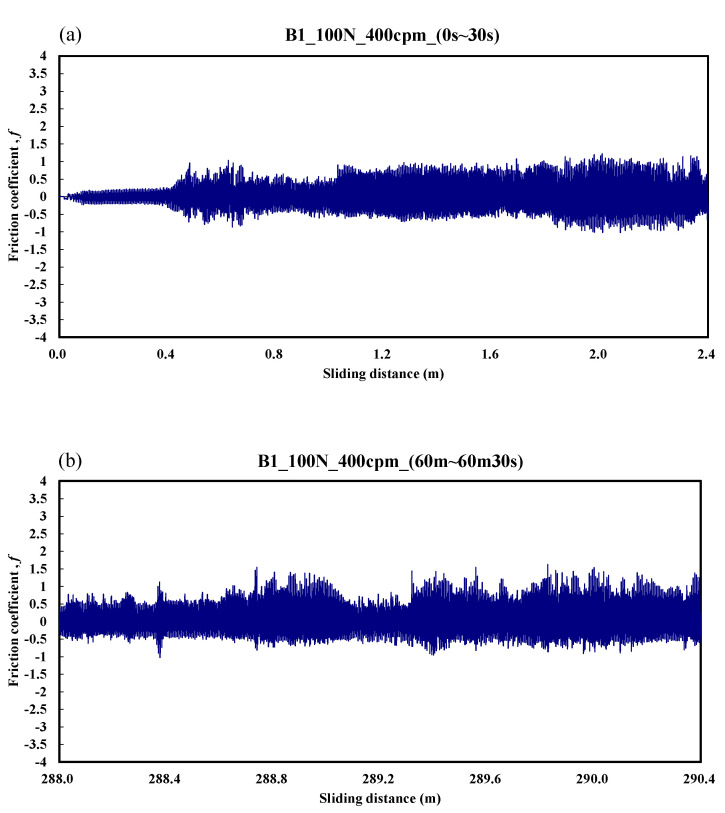
Dynamic responses of the friction coefficient of the B1 specimen. (**a**) sliding distance of 0.0–2.4 m; (**b**) sliding distance of 288.0–290.4 m.

**Figure 9 materials-14-00865-f009:**
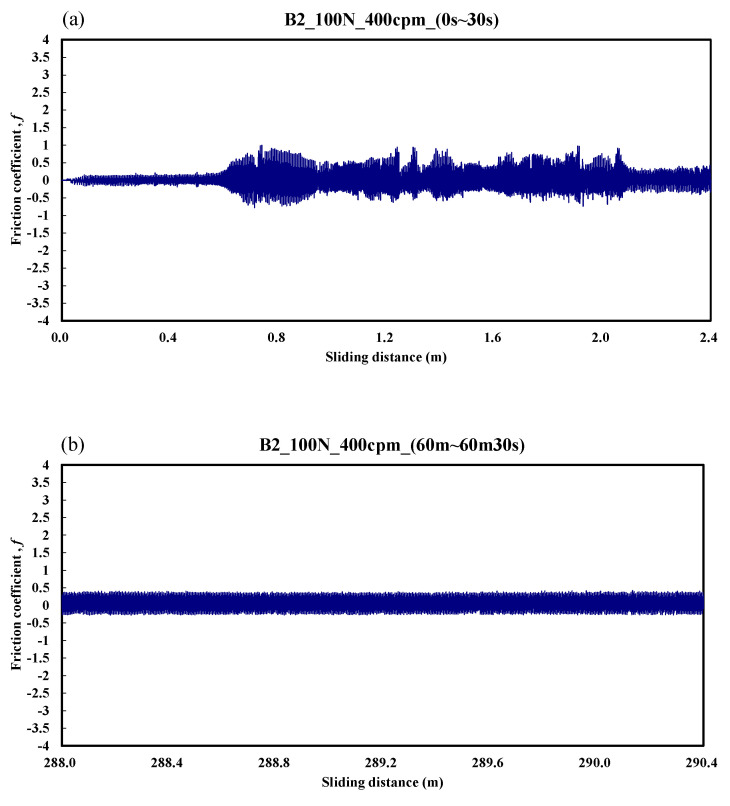
Dynamic responses of the friction coefficient of the B2 specimen. (**a**) sliding distance of 0.0–2.4 m; (**b**) sliding distance of 288.0–290.4 m.

**Figure 10 materials-14-00865-f010:**
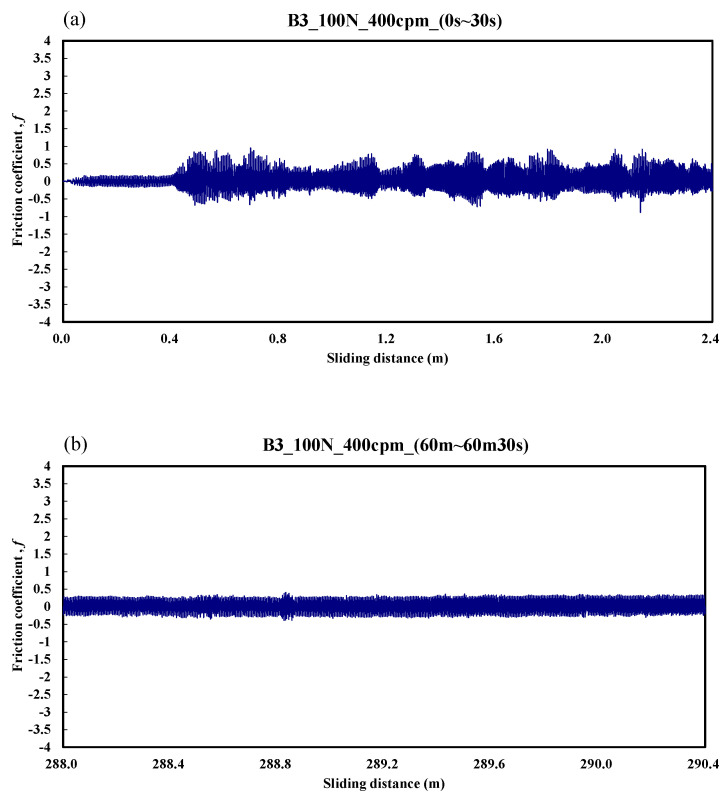
Dynamic responses of the friction coefficient of the B3 specimen. (**a**) sliding distance of 0.0–2.4 m; (**b**) sliding distance of 288.0–290.4 m.

**Figure 11 materials-14-00865-f011:**
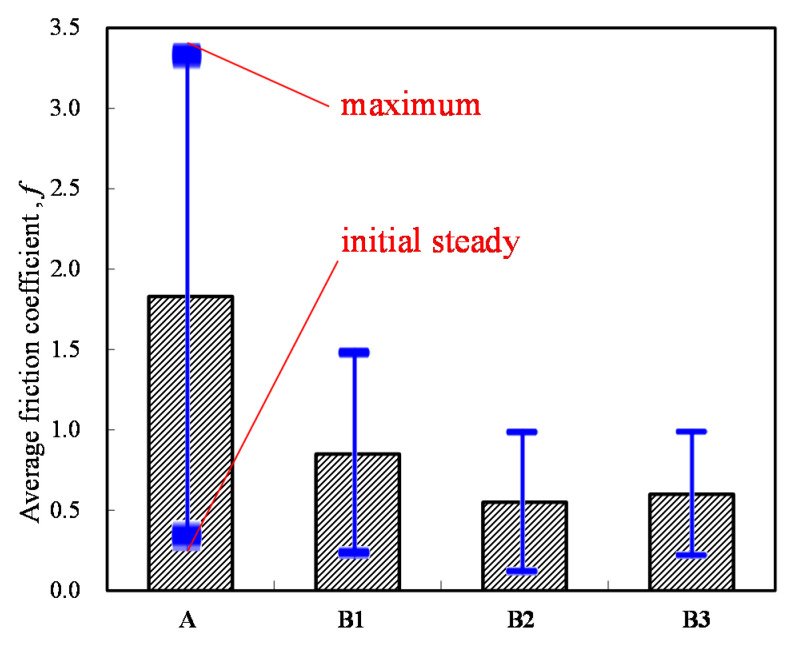
Average friction coefficient of the four specimens.

**Figure 12 materials-14-00865-f012:**
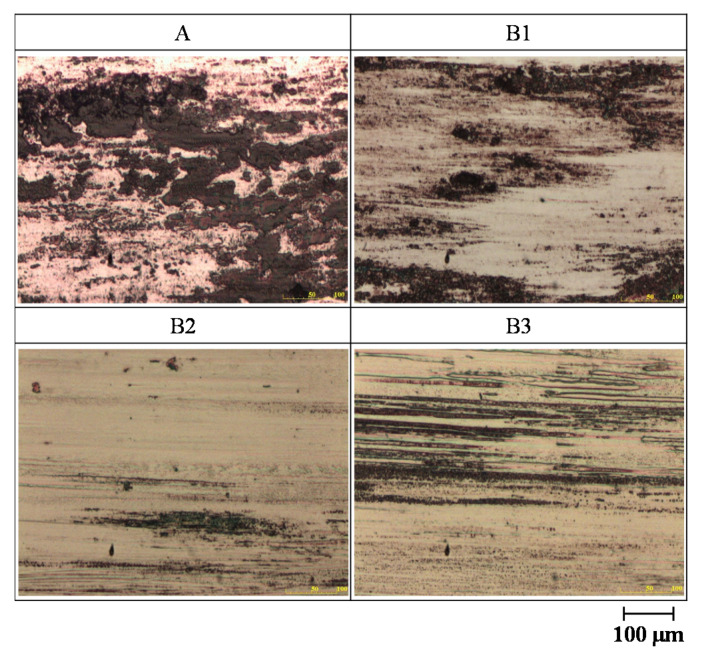
Optical microscope images of the worn surface of the four cylinder specimens: A, B1, B2 and B3.

**Figure 13 materials-14-00865-f013:**
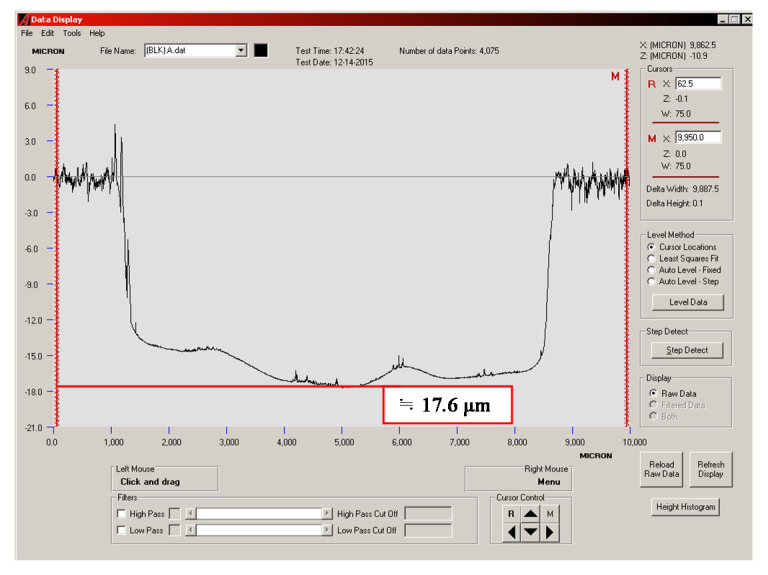
Wear depth of the A specimen.

**Figure 14 materials-14-00865-f014:**
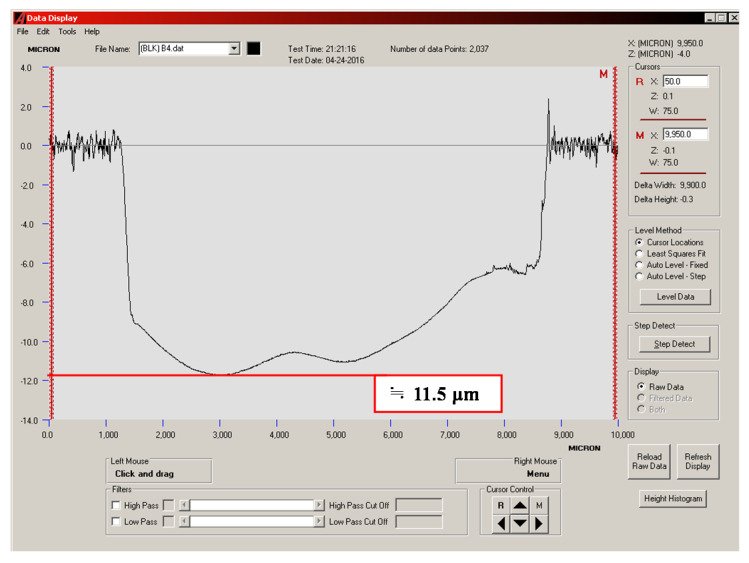
Wear depth of the B1 specimen.

**Figure 15 materials-14-00865-f015:**
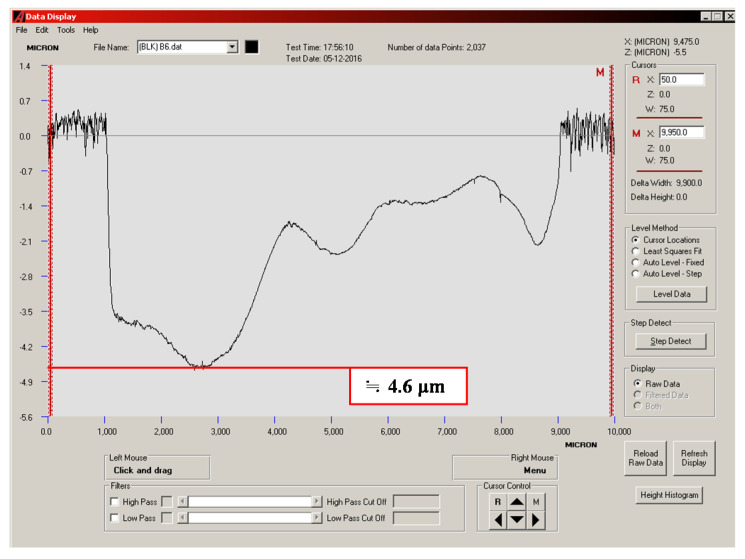
Wear depth of the B2 specimen.

**Figure 16 materials-14-00865-f016:**
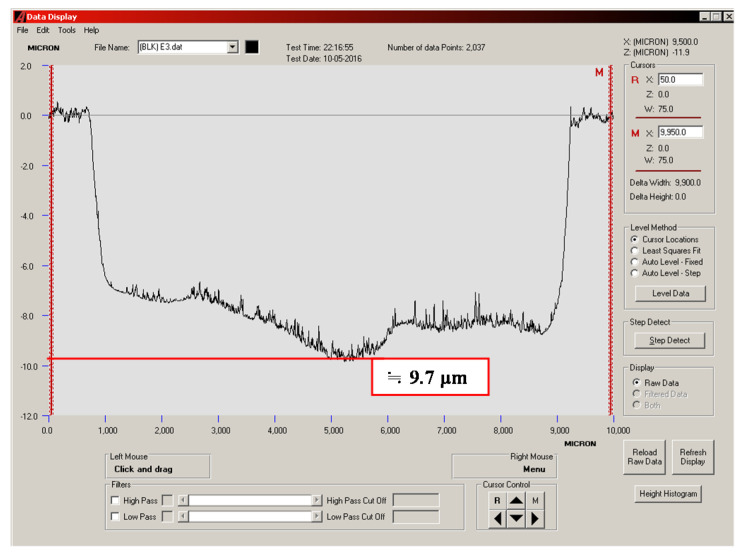
Wear depth of the B3 specimen.

**Figure 17 materials-14-00865-f017:**
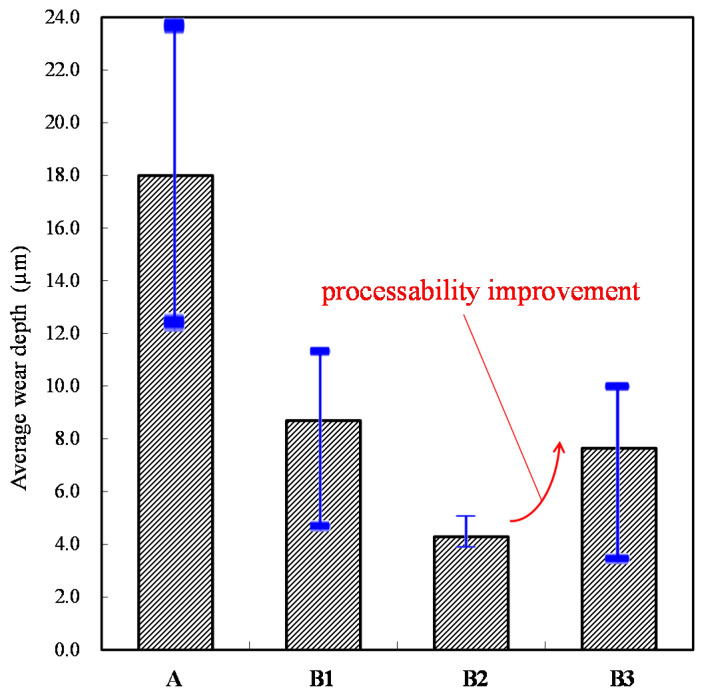
Average wear depths of the four specimens.

**Table 1 materials-14-00865-t001:** Major compositions of the low-carbon alloy steel (wt %).

C	Si	Mn	Ni	Cr	Fe
0.17–0.23	0.15–0.35	0.55–0.95	≤0.25	0.85–1.25	Remaining

**Table 2 materials-14-00865-t002:** Major compositions of the high-carbon chromium alloy steel (wt %).

C	Si	Mn	Cr	Mo	Fe
0.95–1.10	0.15–0.35	≤0.50	1.30–1.60	≤0.08	Remaining

**Table 3 materials-14-00865-t003:** The detailed heat treatment process of the cylinder specimens.

	Material	The Heat Treatment Process	Time of Carburizing or Induction
A	Low-carbon alloy steel	Original material→Carburization→Quenching→Tempering	12 h
B1	High-carbon chromium alloy steel	Medium temperature quenching→ Tempering→Induction (General power)	20 s
B2	High-carbon chromium alloy steel	Medium temperature quenching→Tempering→Induction (General power + Cryogenic treatment)	20 s
B3	High-carbon chromium alloy steel	Medium temperature quenching→Higher temperature tempering→Induction (General power)	20 s

**Table 4 materials-14-00865-t004:** The experimental conditions.

**Ball specimen**	The high-carbon chromium alloy steel (ϕ6.35 mm)
**Cylinder specimen**	The low-carbon alloy steel; the high-carbon chromium alloy steel
**Normal load**	100 N
**Reciprocating speed**	400 cpm
**Stroke**	6 mm
**Friction test time**	60 min 30 s
**Lubrication condition**	Grease

## Data Availability

The data presented in this study are available on request from the corresponding author.

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
