# Peer review of "A Novel Application on the Drive Elements of Using Electrical Contact Resistance and Friction Coefficient for Evaluating Induction Heat Treatment"

_materials, 2021, doi:10.3390/ma14040865_

Round 1
Reviewer 1 Report
This research manuscript has a great potential as denoted in the figures in the result section. However, the introduction and the discussion sections are coarse and does not identify the novelty of this research compare with the previous ones. On top of that, there is a serious issue with the logical English structure in most of the sentences. It is highly recommended a thoroughly extensive editing of English language and style.
Author Response
Reply to Reviewer #1:
- Following the reviewer’s comments, the author will entrust MDPI to provide a thoroughly extensive editing of English language and style.
- For the introduction section, the author would like to add descriptions of quenching, tempering, and cryogenic treatment to show the comparative effect of energy and time consumption.
- For the discussion section, the author would like to add the description and comparison of the composition of the specimens to establish the indicator of the materials.

Reviewer 2 Report
Very interesting, well prepared, and almost correct paper. Results of a presented experimental approach to comparing different steel treatment procedures are interested and promising. The experimental procedure is very interesting and innovative.
However, I have some comments that could be taken into consideration, and the paper should be supplemented with little explanations.
First of all, behaviour and eventually changes in the geometry or surface structure of the ball specimen should be mentioned. Additionally, I did not find any information about the fact, if the ball specimen were the same during whole attempts to experiment, whether if not. For example, using a used ball specimen for further attempts (cylinder specimens) could distort the results significantly. The information about hardness is also very important.
For the second I did not find any information about the value of the temperature of particular treatment steps (quenching, tempering) - this is very important information, and should be indicated more precisely.
For the third, in the discussion (and conclusions) part of the paper, there is not any information about differences in the chemical composition of samples A and B1-B3. Authors suggest that results only depend on the surface treatment methodology, but there are not any explanations and eventually discussion about the possible influence of the chemical composition of the alloys on obtained results. It should be noted that practically two methods of treatment were compared, but unfortunately, they had been used for two different alloys. So in this case we have two variables in general. Even if the hardness of obtained alloys should be indicated, as a more classical indicator, which can be used for evaluation of the proposed test method.
And finally, it should be noticed that authors in the Introduction section are invoking to energy and time consumption during the industrial production process. Of course, the time consumption problem is clearly described using parameter "heat treatment time" and this is indisputable, but in the paper, there is not any information about even if estimated energy consumption, especially that, very high (quenching, tempering) and low (cryogenic) temperature treatment is considered. In both cases, in general, energy consumption is of great importance. If the authors want to focus on it, as indicated in the introduction, at least a rough analysis of these parameters should be performed.
Author Response
Reply to Reviewer #2:
The authors are very grateful for the reviewer’s positive evaluations of this experiment, and will make the following four corrections based on the reviewer’s other comments:
- The ball specimen is a commercial product of SUJ2, and the surface hardness is about HRC 66. Moreover, it is replaced with a new one after each experiment (highlighted the changes in Section 2.2 by using red text). Furthermore, the width of the wear scar of the ball specimen is exactly the same as that of the cylinder specimen, but because the materials of the balls of different experimental groups are exactly the same, the difference is not obvious, so Figure 12 only shows the photo of the wear scar of the cylinder specimen.
- Following the reviewer’s comments, we had added the information about the value of the temperature of particular treatment steps. The appropriate range of quenching temperature is 820~860℃ and lasts about 1hr, and the tempering temperature is about 600~700℃ and lasts about 1.5hr. When the tempering temperature increases by 10°C, the surface hardness will decrease by about HRC 1 (highlighted the changes in Section 2.2 by using red text).
- Following the reviewer’s comments, the author would like to add the description and comparison of the composition of the specimens to establish the indicator of the materials as followings: The original carbon content of A specimen is only about 0.17~0.23 wt%, and the final surface hardness is HRC 59~61 by carburizing for 12 hours. The original carbon content of B1, B2, and B3 is about 0.95~1.10 wt%. After the quenching and tempering treatment, the final surface hardness of about HRC 58~62 can be obtained by 20 seconds of induction (highlighted the improvements in Section 3.4 by using red text).
- Following the reviewer’s comments, the author would like to add the rough analysis of these parameters as followings: The traditional surface hardening treatment includes carburizing for 12hr + quenching for 1hr + tempering for 1.5hr, and the total time is about 14.5hr. The induction heat treatment includes quenching 1hr + tempering 1.5hr + induction for only 20 seconds, and the total time is about 2.5hr. Therefore, the power consumption is reduced by about 5.8 times (highlighted the changes in Introduction by using red text).

Round 2
Reviewer 1 Report
The manuscript have been greatly improved and can be considered for publication.